# Density-based User Representation using Gaussian Process Regression for Multi-interest Personalized Retrieval

**Haolun Wu**[1,2]*   **Ofer Meshi**[3]   **Masrour Zoghi**[3]   **Fernando Diaz**[3]   **Xue Liu**[1,2]
**Craig Boutilier**[3]   **Maryam Karimzadehgan**[3]

[1]McGill University   [2]Mila - Quebec AI Institute   [3]Google Research
haolun.wu@mail.mcgill.ca, xueliu@cs.mcgill.ca, diazf@acm.org
{meshi,mzoghi,cboutilier,maryamk}@google.com

## Abstract

Accurate modeling of the diverse and dynamic interests of users remains a significant challenge in the design of personalized recommender systems. Existing user modeling methods, like single-point and multi-point representations, have limitations w.r.t. accuracy, diversity, and adaptability. To overcome these deficiencies, we introduce *density-based user representations (DURs)*, a novel method that leverages Gaussian process regression (GPR) for effective multi-interest recommendation and retrieval. Our approach, GPR4DUR, exploits DURs to capture user interest variability without manual tuning, incorporates uncertainty-awareness, and scales well to large numbers of users. Experiments using real-world offline datasets confirm the adaptability and efficiency of GPR4DUR, while online experiments with simulated users demonstrate its ability to address the exploration-exploitation trade-off by effectively utilizing model uncertainty.

## 1   Introduction

With the proliferation of online platforms, users have ready access to content, products and services drawn from a vast corpus of candidates. Personalized *recommender systems (RSs)* play a vital role in reducing information overload and helping users navigate this space. It is widely recognized that users rarely have a single intent or interest when interacting with an RS [1, 2, 3]. To enhance personalization, recent work focuses on discovering a user's multiple interests and recommending items that attempt to span their interests [3, 4, 5]. However, this is challenging for two reasons. First, user interests are diverse and dynamic: diversity makes it hard to detect all interests, while their dynamic nature renders determining which user interest is active at any given time quite difficult. Second, it is hard to retrieve items related to niche interests due to the popularity bias [6].

User representation is a fundamental design choice in any RS. The most widely used strategy for user modeling is the *single-point user representation (SUR)*, which uses a single point in an item embedding space to represent the user. The user's affinity for an item is obtained using some distance measure (e.g., inner product, cosine similarity) with the point representing the item. However, SUR can limit the accuracy and diversity of item retrieval [7]; hence, most RSs generally use high-dimensional embedding vectors (with high computation cost).

To address the limitations of SUR, MaxMF [1] adopts a *multi-point user representation (MUR)*, where each user is represented using $K$ points in the embedding space, each reflecting a different "interest". MaxMF uses a constant, uniform $K$ across all users (e.g., $K = 4$), which is somewhat ad hoc and

---

*Work done while doing an internship at Google.

38th Conference on Neural Information Processing Systems (NeurIPS 2024).

restrictive. Subsequent research uses other heuristics [3, 8, 9, 4, 10, 11] or clustering algorithms [5, 2] to determine the number of interests per user. However, these all require the manual choice of $K$ or a specific clustering threshold, limiting the adaptability of MUR methods, since interests generally have high variability across users. Moreover, uncertainty regarding a user's interests is not well-modeled by these methods, diminishing their ability to perform effective online exploration.

To address limitations of SUR and MUR point-based representations, we propose a user representation that emphasizes (i) *adaptability*, adapting to different interest patterns; (ii) *uncertainty-awareness*, modeling uncertainty in assessing user interests; and (iii) *efficiency*, avoiding high-dimensional embeddings. Specifically, we use a *density-based* representation, where the user's preferences are encoded using a *function* over the item embedding space. Under this representation, the relevance score for user-item pairs should be higher in regions of embedding space where a user demonstrated more interest in the corresponding items, and lower in regions where users have shown limited interest. We propose the *density-based user representation (DUR)*, a novel user modeling method which exploits *Gaussian process regression (GPR)* [12, 13], a well-studied Bayesian approach to non-parametric regression using *Gaussian processes (GPs)* to extrapolate from training to test data. GPR has been applied across a wide range of domains, though it has been under-explored in user modeling. Given a sequence of user interactions, GPR predicts the level of a user's interest in any item using its posterior estimates.

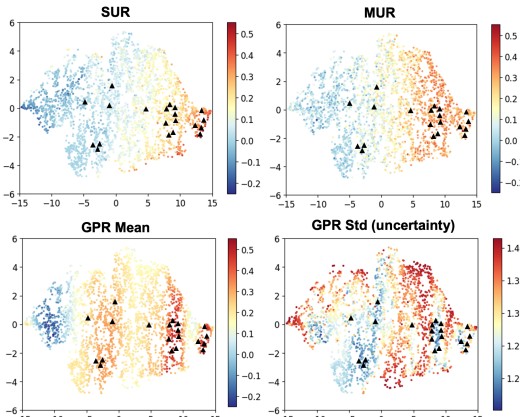

Figure 1: The t-SNE visualization of the prediction score between a picked user to all items in the MovieLens dataset. The score is computed as the inner-product between the user embedding and item embedding. The triangles (▲) indicate the latest 20 items interacted by the user. We use Matrix Factorization (MF) to obtain embeddings in this toy example. As depicted, only density-based method (bottom row) can well capture user interests with uncertainty.

This allows us to maintain a unique personalized GP regressor for each user, effectively capturing their evolving preferences and assessing uncertainty. Top-$N$ item retrieval is performed using bandit algorithms, such as UCB [14] or Thompson sampling [15], based on GPR posterior estimates.

To illustrate, consider Fig. 1, which shows the prediction score (*i.e.*, the inner-product between user and item embeddings) between a user and all movies from the MovieLens 1M dataset [16] (reduced to 2D for visualization purposes). We examine 20 movies from the recent history of a particular user, shown as triangles (▲). We see that these movies lie in several different regions of the embedding space. However, when we fit either SUR or MUR ($K = 4$) models, they fail to capture the user's multiple interests and instead assign high scores only to movies from a single region (Fig. 1, top row). By contrast, we see in Fig. 1 (bottom left) that GPR fits the data well, assigning high values to all regions associated with the user's recent watches (interests). Fig. 1 (bottom right) shows that our approach can also capture uncertainty in our estimates of a user's interests, assigning high uncertainty to regions in embedding space with fewer samples.

Our approach has various desirable properties. First, it adapts to different interest patterns, since the number of interests for any given user is not set manually, but determined by GPR, benefiting from the non-parametric nature of GPs. Second, the Bayesian nature of GPs measures uncertainty with the posterior standard deviation of each item. This supports the incorporation of bandit algorithms in the recommendation and training loop to address the *exploration-exploitation trade-off* in online settings. Finally, our method can effectively retrieve items spanning multiple user interests, including "niche" interests, while using a lower-dimensional embedding relative to SUR and MUR.

To summarize, our work makes the following contributions:

- We develop *GPR4DUR*, a density-based user representation method, for personalized multi-interest retrieval. It is the first use of GPR for user modeling in this setting.

- We propose new evaluation protocols and metrics for multi-interest retrieval that measure the extent to which a model captures a user's multiple interests.

- We conduct comprehensive experiments on real-world offline datasets showing the adaptability and efficiency of GPR4DUR. Online experiments with simulated users show the value of GPR4DUR's uncertainty representation in balancing exploration and exploitation.

## 2   Related Work

Learning high-quality user representations is central to good RS performance. The *single-point user representation (SUR)* is the dominant approach, where a user is captured by a single point in some embedding space [17, 18], for example, as employed by classical [19, 20, 21] and neural [22] collaborative filtering methods. While effective and widely used, SUR cannot reliably capture a user's multiple interests. To address this limitation, the *multi-point user representation (MUR)* [1] has been proposed, where a user is represented by multiple points in embedding space, each corresponding to a different "primary" interest. Selecting a suitable number of points $K$ is critical in MUR. Existing algorithms largely use heuristic methods, e.g., choosing a global constant $K$ for all users [1, 8, 3, 9, 4, 10, 11]. Other methods personalize $K$ by setting it to the logarithm of the number of items with which a user has interacted [5]. More recently, Ward clustering of a user's past items has been proposed, with a user's $K$ determined by the number of such clusters [2]. This too requires manual tuning of clustering thresholds.

At inference time MUR is similar to SUR, computing the inner-product of the user embedding(s) and item embedding. Some methods compute $K$ inner products, one per interest, and use the maximum as the recommendation (and the predicted score for that item) [1]. Others first retrieve the top-$N$ items for each interest ($N \times K$ items), then recommend the top-$N$ items globally [3, 2]. None of these methods capture model uncertainty w.r.t. a user's interests, hence they lack the ability to balance exploration and exploitation in online recommendation in a principled way [23].

Our density-based user representation, and our proposed GPR4DUR, differs from prior work w.r.t. both problem formulation and methodology. Most prior MUR methods focus on *next-item prediction* [3, 11], implicitly assuming a single-stage RS, where the trained model is the main recommendation engine. However, many practical RSs consist of two stages: *candidate selection* (or *retrieval*) followed by *ranking* [24, 25]. This naturally raises the question: ***are the selected candidates diverse enough to cover a user's interests or intents?*** This is especially relevant when a user's dominant interest at the time of recommendation is difficult to discern with high probability; hence, it is important that the ranker have access to a diverse set of candidates that cover the user's *range of potential currently active interests*. In this paper, we focus on this *retrieval task*. As for methodology, almost all prior work uses SUR or MUR point-based representations, [1, 2, 3, 5]—these fail to satisfy all the desiderata outlined in Sec. 1. Instead, we propose DUR, a novel method satisfying these criteria, and, to the best of our knowledge, the first to adopt GPR for user modeling in multi-interest recommendation/retrieval. We frame our solution as a candidate generator to be used in the retrieval phase of an RS. Other candidate generators with different objectives can be used in parallel to ours.

A related non-parametric recommendation approach is $k$-nearest-neighbors (kNN), where users with similar preferences are identified (e.g., in user embedding space), and their ratings generate recommendations (see, e.g., [26]). Our approach differs by not using user embeddings, but fitting a GPR model directly to item embeddings (see Sec. 4.4), allowing for uncertainty in the user model.

## 3   Formulation and Preliminaries

In this section, we outline our notation and multi-interest retrieval problem formulation, and provide some background on GPR, which lies at the core of our DUR method.

**Notation**. We consider a scenario where each item is associated with *category* information (e.g., genre for movies). Denote the set of all *users*, *items*, and *categories* by $\mathcal{U}$, $\mathcal{V}$, and $\mathcal{C}$, respectively. For each $u \in \mathcal{U}$, whose interaction history has length $l_u$, we partition the sequence of items $\mathcal{V}_u$ in $u$'s history into two disjoint lists based on the interaction timestamp (which are monotonic increasing): (i) the *history set* $\mathcal{V}_u^{\mathrm{h}} = [v_{u,1}, v_{u,2}, ..., v_{u,\ell_u}]$ serves as the model input; and (ii) the *holdout set* $\mathcal{V}_u^{\mathrm{d}} = [v_{u,\ell_u+1}, v_{u,\ell_u+2}, ..., v_{u,l_u}]$ is used for evaluation. We define $u$'s *interests* $\mathcal{C}(\mathcal{V}_u)$ to be the *set of categories associated with all items in $u$'s history*. Our notation is summarized in Appendix A.1.

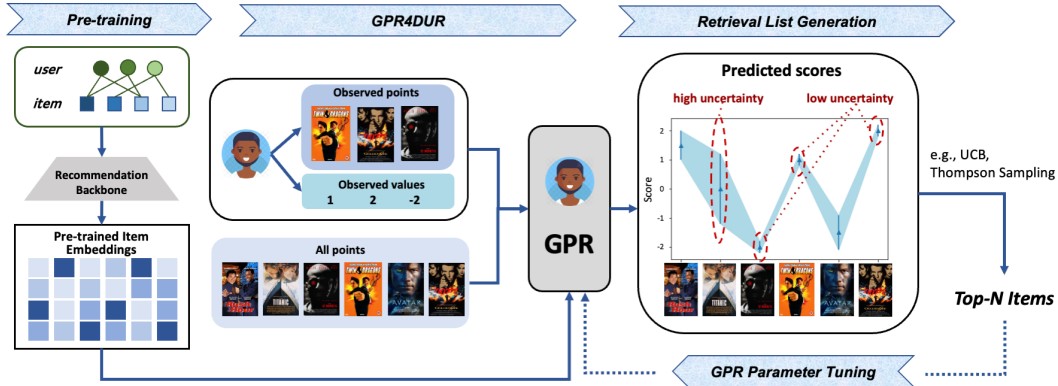

Figure 2: The architecture of GPR4DUR: an example of a movie recommendation for a single user.

**Problem Formulation**. We formulate the *multi-interest retrieval problem* as follows: given $\mathcal{V}_u^h$, we aim to retrieve the top $N$ items $\mathcal{R}_u$ (i.e., $|\mathcal{R}_u| = N$) w.r.t. some *matching metric* connecting $\mathcal{R}_u$ and $\mathcal{V}_u^d$ that measures personalized retrieval performance. We expect $\mathcal{R}_u$ to contain relevant items, and given our focus on multi-interest retrieval, $\mathcal{R}_u$ should ideally cover all categories in $\mathcal{C}(\mathcal{V}_u^d)$.

Our problem is related to *sequential recommendation*, where the input is a sequence of interacted items sorted by the timestamp per user, and the goal is to predict the next item with which the user will interact. However, here we focus on retrieving a set of items that cover a user's diverse interests. We defer the task of generating a precise recommendation list to a downstream ranking model.

**GPR.** The core of our model is *Gaussian process regression (GPR)*. A *Gaussian Process (GP)* is a non-parametric model for regression and classification. GPR models a *distribution over functions*, providing not only function value estimates for any input, but also uncertainty quantification via predictive variance, enhancing robust decision-making and optimization [27].

The key components of the GP are the mean function and the covariance (kernel) function, which capture the prior assumptions about the function's behavior and the relationships between input points, respectively [13]. Let $\mathbf{X} = \{\mathbf{x}_1, \ldots, \mathbf{x}_n\} \in \mathbb{R}^{n \times d}$ be a set of input points and $\mathbf{y} = \{y_1, \ldots, y_n\} \in \mathbb{R}^n$ be the corresponding output values. A GP is defined as: $f \sim \mathcal{GP}(\mu, k)$, where $\mu(\mathbf{x})$ is the mean and $k(\mathbf{x}, \mathbf{x}')$ is the covariance function. The joint distribution of the observation and the output at a new point is derived using a Gaussian distribution, as illustrated in Appendix A.1.

## 4 Methodology

We outline density-based user representation using GPR and its application to multi-interest retrieval.

### 4.1 GPR for Density-based User Representation

We use GPR to construct a novel density-based user representation (DUR) for multi-interest modeling in RSs. Our key insight involves using GPR to learn a DUR, using a user's interaction history, that naturally embodies their diverse interest patterns. For any user $u$, let $\mathbf{V}_u = [\mathbf{v}_{u,1}, \mathbf{v}_{u,2}, ..., \mathbf{v}_{u,l_u}] \in \mathbb{R}^{l_u \times d}$ be the embeddings of all items in their history. This is derived from their interaction list $\mathcal{V}_u$ and an item embedding matrix $\mathbf{V}$ (we describe how to obtain $\mathbf{V}$ in Sec. 4.4). Let $\mathbf{o}_u = [o_{u,v}|v \in \mathcal{V}_u] \in \mathbb{R}^{l_u}$ be the vector of $u$'s *observed interactions* with items in $\mathcal{V}_u$. We employ a GP to model $u$'s interests given the input points $\mathbf{V}_u$ and corresponding observations $\mathbf{o}_u$, $g_u \sim \mathcal{GP}(\mu_u, k_u)$, where, $g_u$, $\mu_u$ and $k_u$ are $u$'s personalized GP regressor, mean, and kernel function, respectively. The joint distribution of observation $\mathbf{o}_u$ and the predicted observation of a novel item $v_*$ is:

$$\begin{bmatrix} \mathbf{o}_u \\ g_u(\mathbf{v}_*) \end{bmatrix} \sim \mathcal{N}\left( \begin{bmatrix} \boldsymbol{\mu}_u(\mathbf{V}_u) \\ \mu_u(\mathbf{v}_*) \end{bmatrix}, \begin{bmatrix} \mathbf{K}(\mathbf{V}_u, \mathbf{V}_u) + \sigma^2 \mathbf{I} & \mathbf{k}(\mathbf{V}_u, \mathbf{v}_*) \\ \mathbf{k}(\mathbf{v}_*, \mathbf{V}_u) & k(\mathbf{v}_*, \mathbf{v}_*) + \sigma_*^2 \end{bmatrix} \right). \tag{1}$$

For simplicity, we assume $\boldsymbol{\mu}_u = \mathbf{0}$ and a commom kernel function and variance across all users. Assuming implicit feedback, we have $\mathbf{o}_u = \mathbf{1}$, i.e., $u$ shows "interest" in all interacted items. Thus,

the posterior (prediction) $g_u(\mathbf{v})$ for any $v \in \mathcal{V}$ is:

$$g_u(\mathbf{v})|\mathbf{o}_u \sim \mathcal{N}(\bar{g}_u, \mathrm{cov}(g_u)), \tag{2}$$

where the GP mean $\bar{g}_u$ and variance $\mathrm{cov}(g_u)$ are:

$$\bar{g}_u = \mathbf{k}(\mathbf{v}, \mathbf{V}_u)[\mathbf{K}(\mathbf{V}_u, \mathbf{V}_u) + \sigma^2\mathbf{I}]^{-1}\mathbf{o}_u, \tag{3}$$

$$\mathrm{cov}(g_u) = k(\mathbf{v},\mathbf{v}) + \sigma^2 - \mathbf{k}(\mathbf{v},\mathbf{V}_u)[\mathbf{K}(\mathbf{V}_u,\mathbf{V}_u) + \sigma^2\mathbf{I}]^{-1}\mathbf{k}(\mathbf{V}_u,\mathbf{v})^T. \tag{4}$$

## 4.2 Retrieval List Generation

After obtaining a DUR $g_u$ for $u \in \mathcal{U}$ using GPR, we generate the *retrieval list* using the posterior $g_u(\mathbf{v})$ over all unobserved items. The top N items with the highest values form our retrieval list. We consider two methods for selection: (i) *Thompson sampling (TS)*, a probabilistic method that selects items based on posterior sampling [15] and (ii) *Upper Confidence Bound (UCB)*, a deterministic method that selects items based on their estimated rewards and uncertainties [14, 28]. The details for the retrieval list generation is shown in Appendix A.2.

## 4.3 GPR Parameter Tuning

The free parameters in our model are the kernel function $k$ and the standard deviation $\sigma$ in Eq. 1. We treat these as hyperparameters of GPR which are optimized using evaluation on a separate holdout set. Specifically, to generate $\mathcal{R}_u$ for user $u$, we fit the GPR model not to the complete interaction history $\mathcal{V}_u$, but to the reduced history $\mathcal{V}_u^{\mathrm{h}}$, using the item embeddings and observed ratings. We assess retrieval performance using specific metrics (see Sec. 5.3) on both $\mathcal{R}_u$ and the holdout set $\mathcal{V}_u^{\mathrm{d}}$. We tune GPR parameters using these criteria (we detail the adjustable parameters in Sec. 5.2).

## 4.4 Item Embedding Pre-training

Following [2], we assume that item embeddings are fixed and precomputed: this ensures rapid computation and real-time updates at serving time. For item embedding pre-training, we use *extreme multi-class classification* [3, 29, 30]: given a training sample $(u_i, v_j)$, we first compute the likelihood of $u_i$ interacting with $v_j$, i.e., $p(v_j|u_i) = \exp(\mathbf{u}_i^\top\mathbf{v}_j)/\sum_{v' \in \mathcal{V}}\exp(\mathbf{u}_i^\top\mathbf{v}')$, where $\mathbf{u}_i$ and $\mathbf{v}_j$ are embeddings of $u_i$ and $v_j$. Our objective is to maximize the log-likelihood of a user interacting with their items. Moreover, we want to ensure that item embeddings align with their categories since categories explicitly indicate a user's interests. We capture item-category information by computing the likelihood that an item belongs to a category in a similar way: $p(c_k|v_j) = \exp(\mathbf{v}_j^\top\mathbf{c}_k)/\sum_{c' \in \mathcal{C}}\exp(\mathbf{v}_j^\top\mathbf{c}')$. The overall objective for pre-training combines the two negative log-likelihoods using a scaling factor $\gamma$:

$$\mathcal{L} = -\sum_{u_i}\sum_{v_j \in \mathcal{V}_{u_i}}\log p(v_j|u_i) - \gamma\sum_{v_j}\sum_{c_k \in \mathcal{C}_{v_j}}\log p(c_k|v_j), \tag{5}$$

where $\mathcal{V}_{u_i}$ is the set of items that $u_i$ interacted with and $\mathcal{C}_{v_j}$ is the set of categories that $v_j$ belongs to. The full item embedding matrix $\mathbf{V}$ is jointly learned and fixed after the pre-training phase. We do not use user and category embeddings after pre-training; other schemes, such as using item co-occurrence information to derive item embeddings without user or category embeddings, are possible. The full architecture for GPR4DUR is depicted in Figure 2.

## 4.5 Computational Complexity

The complexity of GPR is $O((\ell_u)^3)$, where $\ell_u$ is the number of training samples, due to covariance matrix inversion (Eq. 3 and Eq. 4). This inversion is required only once and is manageable with a few thousand examples. If the history length is too large, we either select a representative subset of interactions or focus on the most recent interactions (we adopt this latter approach in our experiments). Inference costs for each test point are $O(\ell_u)$ for mean and $O((\ell_u)^2)$ for variance prediction, thus linear in the number of test points $|\mathcal{V}|$. As shown in Appendix A.7, our method has reasonable computational costs, slightly higher than strong baselines but with better performance on retrieval and ranking, demonstrating its applicability in real-world scenarios. Our approach is best-suited

Table 1: Result comparison on the retrieval task. For the same metric on each dataset, the best is **bold** and the second best is underlined. We use four different symbols to indicate the different categories of methods detailed in Sec. 5.4. Cases where our model significantly outperforms the best baseline, with $p \le 0.01$ according to the paired t-test, are marked with *.

| | Methods | Interest Coverage (IC@k) *The higher the better* ⇑ | | | Interest Relevance (IR@k) *The higher the better* ⇑ | | | Exp. Deviation (ED@k) *The lower the better* ⇓ | | | Tail Exp. Improv. (TEI@k) *The higher the better* ⇑ | | |
|---|---|---|---|---|---|---|---|---|---|---|---|---|---|
| | | k=20 | k=50 | k=100 | k=20 | k=50 | k=100 | k=20 | k=50 | k=100 | k=20 | k=50 | k=100 |
| **Amazon** | ♣ Random | 0.690 | 0.888* | 0.961 | 0.251 | 0.428 | 0.558 | 0.513 | 0.483 | 0.472 | -0.041 | -0.041 | -0.041 |
| | ♣ MostPop | 0.704 | 0.766 | 0.788 | 0.324 | 0.426 | 0.490 | 0.563 | 0.501 | 0.485 | -0.045 | -0.045 | -0.045 |
| | ♦ YoutubeDNN | 0.672 | 0.810 | 0.878 | 0.432 | 0.550* | 0.623 | 0.470* | 0.439 | 0.423 | -0.040 | -0.041 | -0.041 |
| | ♦ GRU4Rec | 0.676 | 0.810 | 0.884 | 0.415 | 0.524 | 0.602 | 0.485 | 0.452 | 0.438 | -0.040 | -0.040 | -0.041 |
| | ♦ BERT4Rec | 0.683 | 0.815 | 0.892 | 0.416 | 0.534 | 0.613 | 0.512 | 0.479 | 0.472 | -0.089 | -0.092 | -0.102 |
| | ♦ gSASRec | 0.672 | 0.823 | 0.899 | 0.418 | 0.535 | 0.618 | 0.506 | 0.472 | 0.469 | -0.066 | -0.069 | -0.069 |
| | ♠ MIND | 0.650 | 0.787 | 0.861 | 0.390 | 0.503 | 0.575 | 0.509 | 0.477 | 0.461 | -0.041 | -0.041 | -0.041 |
| | ♠ ComiRec | 0.656 | 0.785 | 0.861 | 0.399 | 0.510 | 0.595 | 0.492 | 0.453 | 0.432 | -0.040 | -0.040 | -0.040* |
| | ♠ CAMI | 0.640 | 0.710 | 0.833 | 0.373 | 0.483 | 0.521 | 0.522 | 0.493 | 0.473 | -0.071 | -0.088 | -0.080 |
| | ♠ PIMI | 0.705 | 0.821 | 0.897 | 0.415 | 0.535 | 0.622 | 0.497 | 0.462 | 0.441 | -0.092 | -0.091 | -0.144 |
| | ♠ REMI | 0.717* | 0.833 | 0.923 | 0.418 | 0.537 | 0.627* | 0.494 | 0.456 | 0.435 | -0.081 | -0.082 | -0.091 |
| | ♥ **GPR4DUR** | **0.739** | **0.895** | 0.956 | 0.429 | **0.560** | **0.643** | **0.458** | **0.423** | **0.412** | -0.041 | **-0.039** | **-0.039** |
| **MovieLens** | ♣ Random | 0.835 | 0.961 | **0.992** | 0.272 | 0.422 | 0.534 | 0.252 | 0.229 | 0.221 | -0.185 | -0.184 | -0.183 |
| | ♣ MostPop | 0.914* | 0.973* | 0.986 | 0.498 | 0.655 | 0.727 | 0.261 | 0.223 | 0.217 | -0.022 | -0.013* | -0.028* |
| | ♦ YoutubeDNN | 0.879 | 0.938 | 0.974 | 0.722 | 0.846 | 0.873 | 0.250 | 0.229 | 0.218 | -0.017* | -0.031 | -0.051 |
| | ♦ GRU4Rec | 0.832 | 0.939 | 0.976 | 0.646 | 0.784 | 0.863 | 0.228 | 0.226 | 0.195 | -0.070 | -0.076 | -0.084 |
| | ♦ BERT4Rec | 0.883 | 0.852 | 0.963 | 0.732 | 0.847* | 0.883* | 0.271 | 0.258 | 0.256 | -0.072 | -0.087 | -0.103 |
| | ♦ gSASRec | 0.885 | 0.857 | 0.964 | 0.730* | 0.843 | 0.879 | 0.276 | 0.261 | 0.260 | -0.084 | -0.092 | -0.114 |
| | ♠ MIND | 0.869 | 0.951 | 0.981 | 0.653 | 0.786 | 0.863 | 0.245 | 0.223 | 0.212 | -0.049 | -0.058 | -0.073 |
| | ♠ ComiRec | 0.844 | 0.946 | 0.981 | 0.635 | 0.776 | 0.859 | **0.227** | **0.203** | **0.192** | -0.069 | -0.074 | -0.082 |
| | ♠ CAMI | 0.853 | 0.933 | 0.954 | 0.625 | 0.742 | 0.830 | 0.272 | 0.263 | 0.253 | -0.082 | -0.094 | -0.101 |
| | ♠ PIMI | 0.856 | 0.929 | 0.943 | 0.662 | 0.762 | 0.859 | 0.264 | 0.237 | 0.221 | -0.092 | -0.105 | -0.124 |
| | ♠ REMI | 0.861 | 0.930 | 0.947 | 0.718 | 0.792 | 0.868 | 0.270 | 0.252 | 0.230 | -0.102 | -0.113 | -0.156 |
| | ♥ **GPR4DUR** | **0.929** | **0.974** | 0.973 | **0.825** | **0.862** | **0.891** | 0.252 | 0.222 | 0.201 | **-0.011** | **-0.007** | **-0.016** |
| **Taobao** | ♣ Random | 0.302 | 0.563 | 0.873* | 0.232 | 0.416 | 0.527 | 0.493 | 0.425 | 0.333 | -0.059 | -0.049 | -0.031 |
| | ♣ MostPop | 0.342 | 0.583* | 0.863 | 0.362 | 0.439 | 0.643 | 0.512 | 0.437 | 0.343 | -0.076 | -0.050 | -0.036 |
| | ♦ YoutubeDNN | 0.305 | 0.523 | 0.822 | 0.471 | 0.529 | 0.713 | 0.498 | 0.443 | 0.356 | -0.054 | -0.044 | -0.030 |
| | ♦ GRU4Rec | 0.295 | 0.503 | 0.810 | 0.492 | 0.533 | 0.724 | 0.469 | 0.413 | 0.300 | -0.053 | -0.041 | -0.030 |
| | ♦ BERT4Rec | 0.301 | 0.508 | 0.814 | 0.494 | 0.535 | 0.721 | 0.482 | 0.433 | 0.376 | -0.062 | -0.053 | -0.040 |
| | ♦ gSASRec | 0.302 | 0.506 | 0.817 | 0.502 | 0.552 | 0.730 | 0.502 | 0.447 | 0.396 | -0.082 | -0.077 | -0.064 |
| | ♠ MIND | 0.295 | 0.517 | 0.813 | 0.502 | 0.552 | 0.733 | 0.463 | 0.411 | 0.294 | -0.052 | -0.041 | -0.029 |
| | ♠ ComiRec | 0.284 | 0.501 | 0.807 | 0.509 | 0.561 | 0.731 | 0.433 | 0.380 | 0.291 | -0.051 | -0.042 | -0.030 |
| | ♠ CAMI | 0.296 | 0.522 | 0.814 | 0.518* | 0.573* | 0.734* | **0.424** | **0.363** | 0.281* | -0.052 | **-0.039** | -0.027* |
| | ♠ PIMI | 0.343* | 0.579 | 0.860 | 0.516 | 0.566 | 0.729 | 0.498 | 0.447 | 0.328 | -0.089 | -0.052 | -0.035 |
| | ♠ REMI | 0.339 | 0.576 | 0.855 | 0.511 | 0.540 | 0.698 | 0.477 | 0.408 | 0.377 | -0.077 | -0.048 | -0.030 |
| | ♥ **GPR4DUR** | **0.363** | **0.601** | **0.891** | **0.609** | **0.624** | **0.781** | 0.453 | 0.367 | **0.273** | **-0.050** | -0.040 | **-0.023** |

to scenarios with millions or tens of millions of items, when retrieving hundreds for ranking, as is common in real-world RSs (e.g., see [31]). Similarly, in music recommendation, it would be especially useful for defining user interests over artists (a few millions at most per geographical region/language) vs. individual songs. Previous work improves GPR efficiency using Toeplitz and Kronecker structures, achieving $O(\ell_u)$ training and $O(1)$ prediction complexity [32]. Other sublinear approximations partition the item space to find candidates adaptively [33, 34]. While not our focus, it would be straightforward to apply these well-studied methods to our setting.

Finally we note that we do not expect users' interest collections to be highly dynamic; hence, they will not require real-time updates. Our method is best-suited to use as a candidate generator, with the model updated periodically to create a pool of potentially interesting items for users.

# 5 Offline Experiments

We next evaluate our GPR4DUR on three real-world datasets, and compare our DUR technique with other state-of-the-art methods for multi-interest personalized retrieval.

## 5.1 Datasets

We use three widely studied datasets: *Amazon* [35], *MovieLens* [16], and *Taobao* [36]. Following previous work [37, 38], we filter out items with fewer than 10 appearances and users with fewer than 25 interactions to ensure sufficient history for indicating multiple interests. Each interest corresponds to a category: for Amazon, we use the single principal category of each item; for MovieLens, we use 18 movie genres (each movie can belong to multiple genres); for Taobao, we cluster items into 20

categories using Ward clustering on the pretrained item embeddings as in [2]. The same categories are used during training and inference. Dataset statistics are shown in Table 6, and an ablation analysis of various clustering algorithms and numbers of clusters is provided in Table 8 (See Appendix).

## 5.2 Experiment Setup

Most prior work on recommendation and retrieval evaluates model performance on generalization to new items per user. Instead, similar to recent work [39, 3], we assess model performance on the ability to generalize to new users. We split users into disjoint subsets: *training users* ($\mathcal{U}^{\text{train}}$), *validation users* ($\mathcal{U}^{\text{val}}$), and *test users* ($\mathcal{U}^{\text{test}}$) in a ratio of 8:1:1. The last 20% of each user's interaction sequence is treated as a *holdout set* for evaluation, and the first 80% as a *history set* for fitting the GPR model. We cap history length to 60, 160, and 100 for the three datasets, aligning with their average user interactions. For item embedding pre-training, we train the recommendation backbones using the *history set* of all training users and tune parameters based on the performance on the *holdout set* for validation users (details in Appendix A.3). Training is capped at 100,000 iterations with early stopping if validation performance does not improve for 50 successive iterations. We tune GPR hyperparameters by fitting the GP regressor to the *history set* of all training and validation users, then using the *holdout set* to fine-tune parameters (kernel and standard deviation). Metrics are reported on the holdout set for all test users (see Appendix A.5 for hyperparameter sensitivity).

## 5.3 Metrics

Metrics used in prior work on sequential recommendation are not well-suited to assess performance in our multi-interest retrieval task, for several reasons: (i) Conventional metrics, such as *precision* and *recall*, often employed in multi-interest research, do not adequately quantify whether an item list reflects the full range of a user's (multiple) interests. A model may primarily recommend items from a narrow range of highly popular categories and still score high on these metrics while potentially overlooking niche interests. (ii) Metrics like precision and recall are overly stringent, only recognizing items in the recommendation list that appear in the holdout set. We argue that credit should also be given if a similar, though not identical, item is recommended (e.g., *Iron Man 1* instead of *Iron Man 2*). This requires a soft version of these metrics. (iii) Multi-interest retrieval systems should expose users to niche content to address their diverse interests. However, conventional metrics may neglect the item perspective, potentially underserving users with specialized interests. Consequently, we propose the use of the following four metrics that encompass the aforementioned factors.

**Interest-wise Coverage (IC)**  This metric is similar to *subtopic-recall* [40], which directly measures whether the model can comprehensively retrieve all user interests reflected in the holdout set. The higher the value of this metric the better:

$$\text{IC@}k = \frac{1}{|\mathcal{U}^{\text{test}}|} \sum_{u \in \mathcal{U}^{\text{test}}} \frac{|\mathcal{C}(\mathcal{V}_u^{\text{d}}) \cap \mathcal{C}(\mathcal{R}_u^{1:k})|}{|\mathcal{C}(\mathcal{V}_u^{\text{d}})|}. \tag{6}$$

**Interest-wise Relevance (IR)**  To further measure the relevance of retrieved items, we introduce a "soft" recall metric, calculating the maximum cosine similarity between items in the retrieval list and the holdout set within the same category. The motivation for IR is that the success of a retrieval or recommendation list often depends on how satisfying the most relevant item is:

$$\text{IR@}k = \frac{1}{|\mathcal{U}^{\text{test}}|} \sum_{u \in \mathcal{U}^{\text{test}}} \frac{\sum_{c \in \mathcal{C}(\mathcal{V}_u^{\text{d}})} \max_{v_i \in \mathcal{V}_u^{\text{d}}, v_j \in \mathcal{R}_u^{1:k}} S(v_i, v_j)}{|\mathcal{C}(\mathcal{V}_u^{\text{d}})|}, \quad \text{s.t. } \mathcal{C}(v_i) = \mathcal{C}(v_j) = c, \tag{7}$$

where $S(v_i, v_j)$ is the cosine similarity between item $v_i$ and $v_j$. To obtain ground-truth similarities between items, and to mitigate the influence of the chosen pre-trained model, we pretrain the item embeddings using YoutubeDNN [41] with a higher dimension size ($d = 256$) to compute a uniform $\mathbf{S}_{i,j}$ for any backbone. A higher value of this metric is better.

**Exposure Deviation (ED)**  In addition to measuring performance from the user side, we also measure from the item side to test whether exposure of different categories in the retrieval list is close to that in the holdout set. We treat each occurrence of an item category as one unit of exposure,

and compute the normalized exposure vectors $\epsilon_u^*, \epsilon_u^{1:k} \in \mathbb{R}^{|\mathcal{C}|}$ for $u$'s holdout set and retrieval list, respectively. Lower values of this metric are better.

$$\text{ED@}k = \frac{1}{|\mathcal{U}^{\text{test}}|} \sum_{u \in \mathcal{U}^{\text{test}}} ||\epsilon_u^* - \epsilon_u^{1:k}||_2^2, \quad \text{s.t., } \epsilon_{u,c}^* = \frac{\sum_{v \in \mathcal{V}_u^{\text{d}}} \mathbb{1}_{c \in \mathcal{C}(v)}}{\sum_{v \in \mathcal{V}_u^{\text{d}}} |\mathcal{C}(v)|}, \epsilon_{u,c}^{1:k} = \frac{\sum_{v \in \mathcal{R}_u^{1:k}} \mathbb{1}_{c \in \mathcal{C}(v)}}{\sum_{v \in \mathcal{R}_u^{1:k}} |\mathcal{C}(v)|}. \quad (8)$$

**Tail Exposure Improvement (TEI)**   With respect to category exposure, it is crucial to ensure that niche interests are not under-exposed. To evaluate this, we select a subset of the least popular categories and measure their exposure improvement in the retrieval list versus that in the holdout set. A higher value indicates better performance, and a positive value indicates improvement:

$$\text{TEI@}k = \frac{1}{|\mathcal{U}^{\text{test}}|} \sum_{u \in \mathcal{U}^{\text{test}}} \sum_{c \in \mathcal{C}^{\text{tail}}} (\epsilon_{u,c}^{1:k} - \epsilon_{u,c}^*) \mathbb{1}_{\epsilon_{u,c}^* > 0}. \quad (9)$$

Here, $\mathcal{C}^{\text{tail}}$ refers to the set of niche categories (i.e., the last 50% long-tail categories), denoting those niche interests. $\mathbb{1}_{\epsilon_{u,c}^* > 0}$ indicates that we only compute the improvement for categories that appear in the user's holdout set, reflecting their true interests.

## 5.4   Methods Studied

We study the following 12 methods from four categories. *(i) Heuristic* ♣: *Random* recommends random items, *MostPop* recommends the most popular items. *(ii) SUR Methods* ♦: *YoutubeDNN* [41] is a successful deep learning model for industrial recommendation platforms. *GRU4Rec* [30] is the first work to use recurrent neural networks for recommendation. *BERT4Rec* [42] adopts the self-attention mechanism. *gSASRec* [43] is an improvement over SASRec that deploys an increased number of negative samples and a novel loss function. *(iii) MUR Methods* ♠: *MIND* [5] designs

Table 2: Result comparison on the ranking task measured at top-50 across all methods on three datasets.

| | Amazon | | MovieLens | | Taobao | |
|---|---|---|---|---|---|---|
| | Recall | nDCG | Recall | nDCG | Recall | nDCG |
| ♣ Random | 0.836 | 0.848 | 0.685 | 0.716 | 0.838 | 0.866 |
| ♣ MostPop | 0.867 | 0.867 | 0.710 | 0.741 | 0.868 | 0.896 |
| ♦ YoutubeDNN | 0.875 | 0.888 | 0.715 | 0.748 | 0.875 | 0.904 |
| ♦ GRU4Rec | 0.873 | 0.886 | 0.714 | 0.746 | 0.873 | 0.902 |
| ♦ BERT4Rec | 0.881 | 0.892 | 0.723 | 0.758 | 0.883 | 0.914 |
| ♦ gSASRec | 0.880 | 0.894 | 0.725 | 0.760 | 0.884 | 0.915 |
| ♠ MIND | 0.842 | 0.885 | 0.713 | 0.745 | 0.872 | 0.901 |
| ♠ ComiRec | 0.887 | 0.900 | **0.731** | 0.757 | 0.886 | 0.915 |
| ♠ CAMI | 0.892 | 0.905 | 0.725 | 0.761 | 0.891 | 0.920 |
| ♠ PIMI | 0.891 | 0.899 | 0.724 | 0.757 | 0.886 | 0.911 |
| ♠ REMI | 0.900 | 0.902 | 0.728 | 0.760 | 0.889 | 0.916 |
| ♥ **GPR4DUR** | **0.908** | **0.922** | 0.730 | **0.768** | **0.906** | **0.936** |

a multi-interest extractor layer based on the capsule routing. *ComiRec* [3] captures multiple interests from user behavior sequences with a controller for balancing diversity. *CAMI* [11] uses a category-aware multi-interest model to encode users as multiple preference embeddings. *PIMI* [44] models the user representation by considering both the periodicity and interactivity in the item sequence. *REMI* [45] consists of an interest-aware hard negative mining strategy and a routing regularization method. *(iv) DUR Method (Ours)* ♥: *GPR4DUR* uses GPR as a density-based user representation tool for capturing users' diverse interests with uncertainty [2].

**Overall Performance**   Our overall performance comparison addresses two central research questions (*RQs*): whether our proposed method offers superior retrieval performance for users with multiple interests (*RQ1*); and whether it induces appropriate item-sided exposure w.r.t. both popular and niche interests (*RQ2*).

*Performance on the retrieval task*. To assess *RQ1*, we evaluate the effectiveness of the retrieval phase. As shown in Table 1, GPR4DUR outperforms almost all baselines across datasets w.r.t. Interest Coverage (IC@$k$) and Interest Relevance (IR@$k$), demonstrating high degrees of retrieval coverage and relevance. Specifically, on the Amazon dataset, GPR4DUR achieves the highest performance in 4 out of 6 interest metrics, in MovieLens it leads in 5 out of 6 metrics, and in Taobao, it excels across all 6 interest metrics. This shows that GPR4DUR covers a wide range of user interests while simultaneously maintaining high relevance. To assess the statistical significance of our model's performance compared to the best baseline, we performed a paired t-test, which evaluates whether the means of two paired samples differ significantly, ensuring that the observed improvements are not due to random variation.

---

[2]The implementation can be found at https://github.com/haolun-wu/GPR4DUR/.

For *RQ2*, our objective is to ascertain whether item exposure is suitably balanced. We measure this using the Exposure Deviation (ED@$k$) and Tail Exposure Improvement (TEI@$k$) metrics. Lower values of ED@$k$ suggest more satisfying category exposure, while higher values of TEI@$k$ are indicative of enhanced exposure in the long tail item categories. GPR4DUR is highly effective on these metrics, consistently performing well across all datasets in most cases, and validating its ability to provide an optimal level of category exposure. Notably, GPR4DUR achieves the best results in 8 of 9 TEI@$k$ metrics, indicating its superior ability to generate exposure to niche categories/interests. We point out, however, that all TEI@$k$ values are negative, which suggests that none of the methods we tested improve exposure for niche categories relative to the exposure in the user holdout sets. This is explained by the well-known popularity bias inherent in most recommendation methods, and suggests that further work is required on diversification strategies to mitigate such effects.

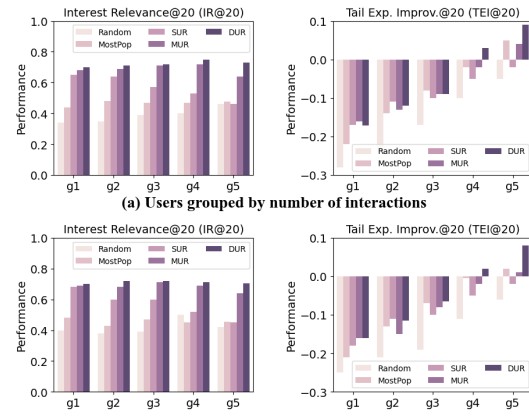

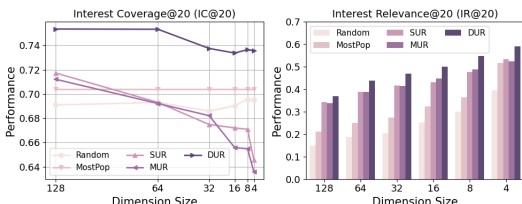

Figure 3: Methods comparison across different user groups on MovieLens. Best viewed in color.

*Performance on the ranking task*. While our primary emphasis is on the retrieval phase, we also present model comparisons for the ranking task. As illustrated in Table 2, GPR4DUR demonstrates competitive performance on the traditional relevance metrics relative to the baseline models. This further validates the strength of our proposed method.

Figure 4: Robustness comparison across different dimension sizes on Amazon. Best viewed in color.

**Performance across User Groups** *(RQ3)* We conduct a fine-grained analysis of GPR4DUR performance on users grouped into quantiles, $g1$ (lowest) through $g5$ (highest), based on the number of interaction or number of interests; results are provided in Fig. 3. Due to space constraints, we show results only for IR@20 and TEI@20 on MovieLens, and only plot the *best* SUR and MUR strategies; DUR denotes our GPR4DUR method. We see that GPR4DUR consistently outperforms the baselines w.r.t. relevance and exposure metrics. This improvement is more pronounced as the number of user interactions (resp., interests) increases. Interestingly, while no model enhances overall exposure of niche interests (see Table 1), GPR4DUR improves niche interest exposure considerably for users with large numbers of interactions (resp., interests); i.e., TEI@20 is positive for $g4$ and $g5$. These observations confirm the effectiveness of GPR4DUR in capturing a user's multiple interests—especially for those with non-trivial histories—and its potential to maintain fair exposure w.r.t. items.

**Robustness to Dimension Size** *(RQ4)* We underscored the importance of efficiency for a good user representation in Sec. 1. To shed light on this, we examine the IC@20 and IR@20 performance of various methods on Amazon. As shown in Fig. 4, GPR4DUR is effective w.r.t. interest coverage even when operating with low-dimensional embeddings (i.e., $d = 8$ and $d = 4$). By contrast, the performance of the SUR and MUR methods degrades as the dimensionality decreases. GPR4DUR consistently outperforms other methods w.r.t. interest relevance across all dimensionalities examined. We note that lower dimensions facilitate higher interest relevance due to increased cosine similarity (Eq. 7), explaining the inverse correlation between IR@20 and dimension size in Figure 4 (right).

## 6   Online Simulation

To demonstrate the efficacy of GPR4DUR in capturing uncertainty in user interests and support exploration, we conduct an online simulation in a synthetic setting, using a specific model of stochastic user behavior to generate responses to recommendations.

**Data Preparation.** We assume $|\mathcal{C}|$ = 10 interest clusters, each represented by a $d$-dim ($d$ = 32) multivariate Gaussian. We randomly select a (user-specific) subset of these interests as the ground-truth interest set for each user. We set $|\mathcal{U}|$ = 1000 and $|\mathcal{V}|$ = 3000, with 300 items in each interest cluster. Each item belongs to a single cluster and its embedding is sampled from the corresponding interest distribution. Each user is modeled by a multi-modal Gaussian, a weighted sum of the corresponding ground-truth interest distributions. To simulate a sequence of item interactions $\mathcal{V}_u$ (i.e., user history), we follow [46], first running a Markov Chain using a predefined user interest transition matrix to obtain the user's interest interactions for $S$ = 10 steps. We do not consider the cold-start problem in this experiment, so we simply recommend one item from each generated cluster to form the user history (i.e., $|\mathcal{V}_u|$ = $S$). The observation $\mathbf{o}_u$ over items in $\mathcal{V}_u$ is set to 1 if the item belongs to a ground-truth cluster, and -1 otherwise.

**GPR Fit and Prediction.** After obtaining item embeddings $\mathbf{V}$, user history $\mathcal{V}_u$, and user observations $\mathbf{o}_u$, we use GPR4DUR to learn a DUR for each user, using the methods described in Sec. 4.

**User History and Observation Update**. Using a predetermined user browsing model, clicked and skipped items are generated and appended to the user interaction history (and the corresponding user observation is likewise updated, 1 for clicked, -1 for skipped). This process continues until a maximum iteration of $T$ = 10 is reached. In this online setting, we adopt the *dependent click model (DCM)* of user browsing behavior, widely used in web search and recommendation [47, 48]. In the DCM, users begin by inspecting the top-ranked item, progressing down the list, engaging with items of interest and deciding to continue or terminate after each viewed item.

**Experiment Results**. We compare different recommendation policies by assessing various metrics at each iteration; due to space limitations, we report results only for interest coverage, see Table 3. Specifically, we compute a "cumulative" version of interest coverage by reporting the interest coverage averaged across all users on all previously recommended items prior to the current iteration. Our goal is to test whether policies that use uncertainty models outperform those that do not, specifically, whether such policies can exploit the inherent uncertainty representation of user interests offered by GPR4DUR.

Table 3: Comparison between different policies in online setting. The reported values are the interest coverage averaged across all users on all *cumulative* recommended items up to each iteration. The highest value per column is bold.

| Policy | $t$=1 | $t$=2 | $t$=3 | $t$=4 | $t$=5 | $t$=6 | $t$=7 | $t$=8 | $t$=9 | $t$=10 |
|---|---|---|---|---|---|---|---|---|---|---|
| Random | 0.29 | 0.49 | 0.64 | 0.74 | 0.82 | 0.87 | 0.91 | 0.91 | 0.92 | 0.93 |
| Greedy | 0.71 | 0.88 | 0.90 | 0.90 | 0.91 | 0.91 | 0.91 | 0.92 | 0.92 | 0.92 |
| UCB ($\beta$=1) | 0.71 | **0.89** | **0.91** | **0.91** | **0.92** | **0.92** | **0.92** | **0.93** | 0.94 | 0.95 |
| UCB ($\beta$=5) | **0.72** | 0.88 | 0.90 | 0.90 | 0.91 | 0.91 | **0.92** | 0.92 | 0.93 | 0.94 |
| Thompson | 0.29 | 0.50 | 0.65 | 0.75 | 0.82 | 0.89 | 0.91 | 0.92 | **0.95** | **0.98** |

We assume each policy recommends the top-10 items to each user at each iteration. Table 3 shows that methods using uncertainty (the bottom three rows) fairly reliably outperform those that do not (the top two rows, with *Greedy* being UCB with $\beta$ = 0, where $\beta$ is the scaling factor of the variance term in UCB). The observation confirms the benefit of using GPR4DUR to explicitly model uncertainty in a recommender's estimates of a user's interests and to use this to drive exploration.

# 7 Conclusion and Discussion

In this paper, we introduced a density-based user representation model, GPR4DUR, marking the first application of Gaussian process regression for user modeling in multi-interest retrieval. This innovative approach inherently captures dynamic user interests, provides uncertainty-awareness, and proves to be more dimension-efficient than traditional point-based methods. We also establish a new evaluation protocol, developing new metrics specifically tailored to multi-interest retrieval tasks, filling a gap in the current evaluation landscape. Offline experiments validate the adaptability and efficiency of GPR4DUR, and demonstrate significant benefits relative to existing state-of-the-art models. Online simulations further highlight GPR4DUR's ability to drive user interest exploration by recommendation algorithms that can effectively leverage model uncertainty. The future work and broader impacts of this study are discussed in Appendix A.9 and Appendix A.10, respectively.

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

# A  Appendix

## A.1  Notations and Preliminaries

### A.1.1  Notations

Table 4: Description of Notation.

| Notation | Description |
|---|---|
| $\mathcal{U}, \mathcal{V}, \mathcal{C}$ | The set of all users, items, and (item) categories. |
| $\mathbf{U}, \mathbf{V}, \mathbf{C}$ | Full embedding matrix of users, items, and categories. |
| $\mathcal{V}_u$ | The user $u$'s interaction history. |
| $\mathcal{V}_u^{\mathrm{h}}, \mathcal{V}_u^{\mathrm{d}}$ | The history set and holdout set partitioned from $\mathcal{V}_u$. |
| $\mathbf{o}_u$ | The observed rating scores of $u$ on items in $\mathcal{V}_u$. |
| $t_{u,v}$ | The time step when $u$ interacted with $v$. |
| $l_u$ | The length of $\mathcal{V}_u$. |
| $\ell_u$ | The length of user history for model input. |
| $\mathbf{V}_u$ | Item embeddings for items in $\mathcal{V}_u$. |
| $\mathcal{R}_u$ | The list of retrieved items to $u$. |
| $\mathcal{C}(\cdot)$ | The set of categories of all items in the input sequence. |

### A.1.2  Preliminaries on Gaussian Process Regression

Let $\mathbf{X} = \{\mathbf{x}_1, \ldots, \mathbf{x}_n\} \in \mathbb{R}^{n \times d}$ be a set of input points and $\mathbf{y} = \{y_1, \ldots, y_n\} \in \mathbb{R}^n$ be the corresponding output values. A GP is defined as:

$$f \sim \mathcal{GP}(\mu, k), \tag{10}$$

where $\mu(\mathbf{x})$ is the mean function and $k(\mathbf{x}, \mathbf{x}')$ is the covariance function (kernel). Given a new point $\mathbf{x}_*$, the joint distribution of the observed outputs and the output at the new point is given by:

$$\begin{bmatrix} \mathbf{y} \\ f(\mathbf{x}_*) \end{bmatrix} \sim \mathcal{N}\left( \begin{bmatrix} \boldsymbol{\mu}(\mathbf{X}) \\ \mu(\mathbf{x}_*) \end{bmatrix}, \begin{bmatrix} \mathbf{K}(\mathbf{X}, \mathbf{X}) + \sigma^2 \mathbf{I} & \mathbf{k}(\mathbf{X}, \mathbf{x}_*) \\ \mathbf{k}(\mathbf{x}_*, \mathbf{X}) & k(\mathbf{x}_*, \mathbf{x}_*) + \sigma_*^2 \end{bmatrix} \right), \tag{11}$$

where $\boldsymbol{\mu}(\mathbf{X})$ is the vector of mean values for the observed data points, $\mathbf{K}(\mathbf{X}, \mathbf{X})$ is the covariance matrix for the observed data points, $\mathbf{k}(\mathbf{X}, \mathbf{x}_*)$ is the vector of covariances between the observed data points and the new input point, and $\sigma^2$ and $\sigma_*^2$ are the noise variances. Without prior observations, $\boldsymbol{\mu}$ is generally set as $\mathbf{0}$.

The conditional distribution of $f(\mathbf{x}_*)$ given the observed data is:

$$f(\mathbf{x}_*)|\mathbf{y} \sim \mathcal{N}(\bar{f}_*, \mathrm{cov}(f_*)), \tag{12}$$

with the predictive mean and covariance given by:

$$\bar{f}_* = \mu(\mathbf{x}_*) + \mathbf{k}(\mathbf{x}_*, \mathbf{X})[\mathbf{K}(\mathbf{X}, \mathbf{X}) + \sigma^2 \mathbf{I}]^{-1}(\mathbf{y} - \boldsymbol{\mu}), \tag{13}$$

$$\mathrm{cov}(f_*) = k(\mathbf{x}_*, \mathbf{x}_*) + \sigma_*^2 - \mathbf{k}(\mathbf{x}_*, \mathbf{X})[\mathbf{K}(\mathbf{X}, \mathbf{X}) + \sigma^2 \mathbf{I}]^{-1}\mathbf{k}(\mathbf{X}, \mathbf{x}_*)^T. \tag{14}$$

Fig. 5 presents a visual illustration of GPR. The true underlying function is depicted in red, which is the function we aim to approximate through GPR. The observations, depicted as black crosses, represent known data points. As expected with GPR, where data points are observed, the uncertainty (represented by the shaded region) is minimal, signifying high confidence in predictions at those locations. On the other hand, in areas without observations, the uncertainty increases, reflecting less confidence in the model's predictions. The two dashed lines represent samples from the GP posterior. Around observed points, these sampled functions adhere closely to the actual data, representing the power and flexibility of GPR in modeling intricate patterns based on sparse data.

## A.2  Details for Retrieval List Generation

After obtaining a DUR $g_u$ for $u \in \mathcal{U}$ using GPR, we generate the retrieval list using the posterior $g_u(\mathbf{v})$ over all unobserved items. The top-N items with the highest values are selected as our retrieval list. We consider two methods for estimating these values.

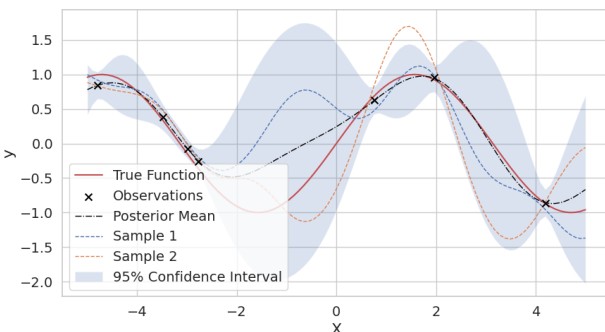

Figure 5: Illustration of Gaussian Process Regression in 1D. The true function is shown in red, observations are marked with black crosses, and the dashed lines represent two samples from the GP posterior. The dash-dot line represents the posterior mean, while the shaded region indicates the 95% confidence interval, showcasing the uncertainty associated with the GP predictions.

The first is *Thompson sampling (TS)*, a probabilistic method that selects items based on posterior sampling [15]. For each item $v$ in a set, we generate a sample from the posterior $g_u(\mathbf{v})$, and rank the items using their sampled values. A key advantage of TS is its ability to balance the trade-off between exploration and exploitation, improving the diversity of the recommendation list. The sampling and selection process is:

$$s_{u,v} \sim g_u(\mathbf{v}), \quad \forall v \notin \mathcal{I}_u, \tag{15}$$
$$\mathcal{R}_u = \text{Top-N}(s_{u,v}), \tag{16}$$

where $s_{u,v}$ is the value for item $v$ from user $u$'s sampled function, and $\mathcal{R}_u$ is the final list of retrieved items for user $u$.

The second method is *Upper Confidence Bound (UCB)*, a deterministic method that selects items based on their estimated rewards and uncertainties [14, 28]. For each item $v$, we compute its upper confidence bound by adding the mean and a confidence interval derived from the variance of the posterior $g_u(\mathbf{v})$; items are ranked using these upper bounds. Unlike TS, UCB tends to aggressively promote items with a high degree of posterior uncertainty, giving a different flavor of diversity in the recommendation list. The selection process is:

$$b_{u,v} = \bar{g}_u(\mathbf{v}) + \beta \cdot \sqrt{\text{var}[g_u(\mathbf{v})]}, \quad \forall v \notin \mathcal{I}_u, \tag{17}$$
$$\mathcal{R}_u = \text{Top-N}(b_{u,v}), \tag{18}$$

where $b_{u,v}$ is the upper confidence bound for $v$ w.r.t. $u$'s posterior, and $\beta$ is a hyper-parameter that adjusts the exploration-exploitation trade-off. $\mathcal{R}_u$ is the final retrieval list.

### A.3 Pretraining Strategies

**Necessity of Pretraining Item Embeddings with Auxiliary Loss on Categories**. Without augmenting the pre-training loss with categorical information as in Eq. 5, the resulting pre-trained embeddings do not respect category consistency. To illustrate this, we compare pre-trained embeddings with and without categorical information (on MovieLens) and show the results in Fig. 6.

We observe that, without categorical constraints, the learned embeddings do not align with the categories. There are still overlap between categories (genres) since each movie may belong to multiple genres. Hence, incorporating an auxiliary training loss based on item categories is essential for interest-coverage analysis based on the embeddings.

**Sensitivity to Different Pretraining Strategies**. In this work, we use pretrained embeddings for our multi-interest user representation. To ensure a fair comparison, we use the same pretrained embeddings in all other models we compared. To test the impact of the embeddings on performance, we add experiments with other pretraining settings. Specifically, we obtain embeddings using matrix factorization (MF) and using YoutubeDNN with varying history lengths. We use the Amazon dataset as an example and measure performance at top-50. The observation is similar on the other two datasets. Other experiment settings are the same as presented in the paper. We use the learned

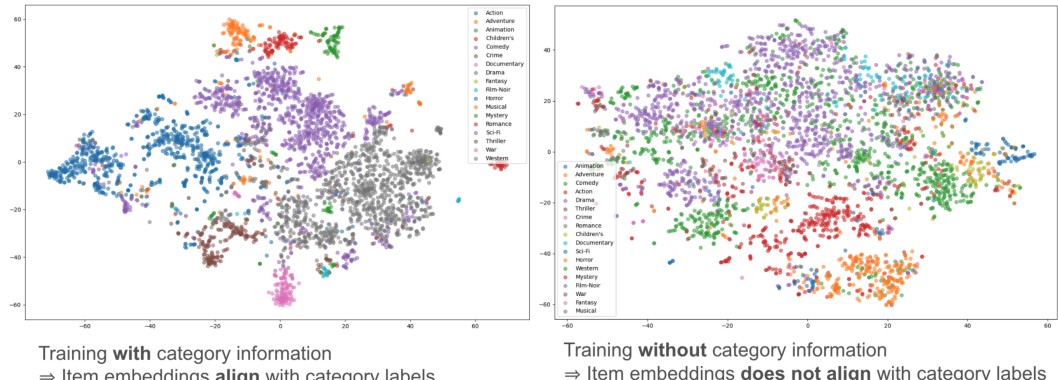

Training **with** category information
⇒ Item embeddings **align** with category labels

Training **without** category information
⇒ Item embeddings **does not align** with category labels

Figure 6: The t-SNE plot for pretrained item embeddings on the MovieLens dataset with and without using auxiliary loss on category information.

Table 5: Performance comparison of different pretraining settings on various metrics. ⇑ indicates the higher the better; ⇓ indicates the lower the better.

| Pretraining setting | | Retrieval Task | | | | Ranking Task | |
| --- | --- | --- | --- | --- | --- | --- | --- |
| | | IC@50 ⇑ | IR@50 ⇑ | ED@50 ⇓ | TEI@50 ⇑ | Recall@50 ⇑ | nDCG@50 ⇑ |
| MF | ComiRec | 0.684 | 0.417 | 0.523 | -0.124 | 0.865 | 0.851 |
| | **GPR4DUR** | **0.887** | **0.527** | **0.501** | **-0.052** | **0.882** | **0.871** |
| YoutubeDNN (max user history=20) | ComiRec | 0.709 | 0.460 | 0.502 | -0.071 | 0.878 | 0.870 |
| | **GPR4DUR** | **0.893** | **0.552** | **0.426** | **-0.056** | **0.892** | **0.913** |
| YoutubeDNN (max user history=50) | ComiRec | 0.710 | 0.463 | 0.493 | -0.040 | 0.887 | 0.900 |
| | **GPR4DUR** | **0.895** | **0.560** | **0.423** | **-0.039** | **0.908** | **0.922** |
| YoutubeDNN (max user history=100) | ComiRec | 0.711 | 0.465 | 0.490 | -0.039 | 0.890 | 0.901 |
| | **GPR4DUR** | **0.896** | **0.562** | **0.423** | **-0.037** | **0.910** | **0.925** |

embeddings with both ComiRec (one of the strongest baselines) and our GPR4DUR. The results, shown in Table 5, demonstrate that embeddings from YoutubeDNN outperform those from MF and that longer history leads to better performance. These results highlight the effectiveness and robustness of our proposed method across different pretraining settings.

## A.4 Dataset Statistics

The statistics of the three datasets, *Amazon* [35], *MovieLens* [16], and *Taobao*[3], is shown in Table 6. We report the number of users, number of items, number of total interactions, and density of the three datasets, respectively.

## A.5 Hyperparameter Settings and Sensitivity Analysis for GPR Parameter Tuning

In short, we constantly use the radial basis function (RBF) kernel on different datasets due to its superior performance, where $\sigma$ is unique for each dataset.

The kernel function, $k$, is tuned from the cosine similarity kernel and the RBF kernel. For the RBF kernel, the standard deviation $\sigma$ is tuned from $\{1e^{-2}, 1e^{-1}, 1, 10, 100\}$. For each dataset, we report one metric on the retrieval task and one metric on the ranking task, both evaluated at the top-50. The trend on other metrics is similar.

As shown in Table 7, we observe that the RBF kernel outperforms the cosine similarity kernel across various datasets and metrics because it better captures non-linear relationships and offers flexibility by mapping inputs into an infinite-dimensional space. Additionally, the $\sigma$ parameter in the RBF kernel influences model performance—a larger $\sigma$ yields smoother functions for better generalization over distances, while a smaller $\sigma$ detects finer data variations. Dataset-specific optimal $\sigma$ values correlate with data sparsity. MovieLens's high density demands a smaller $\sigma$ to avoid overfitting due

---

[3]https://tianchi.aliyun.com/dataset/649?lang=en-us

Table 6: The statistics of datasets.

|           | # User  | # Item  | # Interac. | Density |
|-----------|---------|---------|------------|---------|
| Amazon    | 6,223   | 32,830  | 4M         | 0.18%   |
| MovieLens | 123,002 | 12,532  | 20M        | 1.27%   |
| Taobao    | 756,892 | 570,350 | 70M        | 0.01%   |

Table 7: Hyperparameter Settings and Sensitivity Analysis for GPR Parameter Tuning.

| GPR Parameters | Amazon | | MovieLens | | Taobao | |
|----------------|---------|-----------|---------|-----------|---------|-----------|
|                | IC@50 ⇑ | Recall@50 ⇑ | IC@50 ⇑ | Recall@50 ⇑ | IC@50 ⇑ | Recall@50 ⇑ |
| Cosine | 0.809 | 0.874 | 0.940 | 0.710 | 0.518 | 0.870 |
| RBF ($\sigma = 1e^{-2}$) | 0.822 | 0.877 | 0.963 | 0.727 | 0.536 | 0.872 |
| RBF ($\sigma = 1e^{-1}$) | 0.883 | 0.892 | **0.974** | **0.730** | 0.587 | 0.885 |
| RBF ($\sigma = 1$) | **0.895** | **0.908** | 0.957 | 0.725 | 0.591 | 0.887 |
| RBF ($\sigma = 10$) | 0.887 | 0.889 | 0.954 | 0.723 | **0.601** | **0.906** |
| RBF ($\sigma = 100$) | 0.890 | 0.890 | 0.954 | 0.721 | 0.592 | 0.899 |

to capturing detailed variations. In contrast, Taobao's low density necessitates a larger $\sigma$ for greater generalization, with Amazon's intermediate sparsity requiring a moderate $\sigma$ for optimal performance.

### A.6 Ablation Analysis on Different Clustering Options

In this section, we provide the ablation analysis on using various clustering algorithms and different number of clusters on the Taobao dataset. Following paper [49], we have selected four distinct clustering algorithms for our comparative analysis: (i) *Ward* [50]: This hierarchical clustering technique focuses on minimizing the variance within each cluster, effectively reducing the overall sum of squared distances across all clusters. (ii) *K-Means* [51]: An iterative clustering algorithm that aims to minimize the sum of squared distances from each data point to the centroid of its assigned cluster, thereby reducing in-cluster variance. (iii) *Spectral Clustering* [52]: This method performs clustering based on the eigenvectors of the normalized Laplacian, which is computed from the affinity matrix, facilitating the division based on the graph's inherent structure. (iv) *BIRCH* [53]: A hierarchical clustering approach that efficiently processes large datasets by incrementally constructing a Clustering Feature Tree, which groups data points based on their proximity and other features.

A notable deviation from paper [49] is in our handling of categories (interests) for training and inference phases; we maintain consistency in categories across both phases, in contrast to their separate categorization. We further refine our analysis by tuning the number of clusters from 10, 20, and 50.

**Result and analysis.** The results presented in the Table 8 indicate a nuanced understanding of the impact of clustering algorithms and the number of clusters on retrieval and ranking tasks in information retrieval systems. Across the board, the differences among the clustering algorithms—Ward, K-Means, Spectral, and BIRCH—are relatively small, suggesting that the choice of clustering algorithm might not be as critical as the configuration of the number of clusters within these algorithms for the tasks at hand.

Notably, as the number of clusters increases from 10 to 20 and then to 50, there is a general trend of improvement in performance metrics. This improvement could be attributed to the algorithms' ability to capture more fine-grained user interests through a larger number of clusters, thereby enhancing both retrieval accuracy and ranking precision.

However, the marginal differences observed between the configurations of 20 and 50 clusters across all metrics indicate diminishing returns on further increasing the number of clusters beyond 20. This observation suggests that while increasing the number of clusters to 20 contributes to significant improvements, escalating to 50 clusters does not yield proportionally higher gains. Consequently, this study opts for a configuration of 20 clusters as the optimal balance between performance improvement and computational efficiency. The relatively stable standard deviations across different configurations

Table 8: Result comparison on Taobao dataset using different cluster options. The Interest Coverage (IC) metric is omitted from this report as its value is significantly influenced by the chosen number of clusters, rendering it less relevant for comparative purposes. All reported metrics are evaluated at the top-50. ⇑ indicates the higher the better; ⇓ indicates the lower the better.

| | Num of clusters | Retrieval Task | | | Ranking Task | |
|---|---|---|---|---|---|---|
| | | IR@50 ⇑ | ED@50 ⇓ | TEI@50 ⇑ | Recall@50 ⇑ | nDCG@50 ⇑ |
| Ward | 10 | $0.602 \pm 0.012$ | $0.385 \pm 0.011$ | $-0.055 \pm 0.001$ | $0.892 \pm 0.013$ | $0.920 \pm 0.012$ |
| | 20 | $\underline{0.624 \pm 0.011}$ | $\underline{0.367 \pm 0.011}$ | $\mathbf{-0.040 \pm 0.001}$ | $\mathbf{0.906 \pm 0.012}$ | $\underline{0.936 \pm 0.011}$ |
| | 50 | $\mathbf{0.625 \pm 0.011}$ | $\mathbf{0.365 \pm 0.011}$ | $\underline{-0.041 \pm 0.001}$ | $\mathbf{0.906 \pm 0.012}$ | $\mathbf{0.938 \pm 0.011}$ |
| K-Means | 10 | $0.590 \pm 0.013$ | $0.392 \pm 0.011$ | $-0.060 \pm 0.001$ | $0.885 \pm 0.013$ | $0.912 \pm 0.012$ |
| | 20 | $\underline{0.615 \pm 0.012}$ | $\underline{0.370 \pm 0.011}$ | $\mathbf{-0.045 \pm 0.001}$ | $\underline{0.900 \pm 0.012}$ | $\underline{0.928 \pm 0.012}$ |
| | 50 | $\mathbf{0.620 \pm 0.011}$ | $\mathbf{0.368 \pm 0.011}$ | $\underline{-0.046 \pm 0.001}$ | $\mathbf{0.904 \pm 0.012}$ | $\mathbf{0.930 \pm 0.011}$ |
| Spectral | 10 | $0.605 \pm 0.012$ | $0.382 \pm 0.011$ | $-0.053 \pm 0.001$ | $0.895 \pm 0.013$ | $0.923 \pm 0.012$ |
| | 20 | $\mathbf{0.625 \pm 0.011}$ | $\underline{0.366 \pm 0.011}$ | $\mathbf{-0.039 \pm 0.001}$ | $\underline{0.908 \pm 0.012}$ | $\mathbf{0.937 \pm 0.011}$ |
| | 50 | $\underline{0.624 \pm 0.011}$ | $\mathbf{0.364 \pm 0.011}$ | $\mathbf{-0.039 \pm 0.001}$ | $\mathbf{0.909 \pm 0.011}$ | $\underline{0.936 \pm 0.012}$ |
| BIRCH | 10 | $0.580 \pm 0.013$ | $0.395 \pm 0.012$ | $-0.065 \pm 0.002$ | $0.880 \pm 0.014$ | $0.910 \pm 0.013$ |
| | 20 | $\mathbf{0.610 \pm 0.012}$ | $\underline{0.372 \pm 0.011}$ | $\mathbf{-0.050 \pm 0.001}$ | $0.899 \pm 0.012$ | $0.926 \pm 0.012$ |
| | 50 | $\mathbf{0.610 \pm 0.011}$ | $\mathbf{0.370 \pm 0.011}$ | $\underline{-0.051 \pm 0.001}$ | $\mathbf{0.902 \pm 0.012}$ | $\mathbf{0.928 \pm 0.011}$ |

Table 9: Latency and performance comparison across models on the Taobao dataset, which contains 570K items. Training and inference latencies are measured in millisecond (ms). Standard deviation is shown in parentheses. We see that the cost of inference is on par with other methods.

| | YoutubeDNN | GRU4Rec | BERT4Rec | gSASRec | MIND | ComiRec | CAMI | PIMI | REMI | GPR4DUR (Ours) |
|---|---|---|---|---|---|---|---|---|---|---|
| Train (std) | 153.08 (1.93) | 262.28 (1.20) | 926.36 (82.39) | 492.10 (25.63) | 552.37 (15.50) | 575.51 (67.32) | 602.38 (84.31) | 611.36 (73.02) | 732.16 (32.97) | 932.00 (16.27) |
| Inference (std) | 1.03 (0.76) | 1.44 (0.81) | 49.28 (21.47) | 20.05 (42.07) | 18.62 (52.25) | 18.89 (54.57) | 38.24 (43.18) | 40.57 (39.65) | 49.27 (56.33) | 63.67 (26.38) |
| IR@50 | 0.529 | 0.533 | 0.557 | 0.552 | 0.552 | 0.561 | 0.573 | 0.566 | 0.540 | **0.624** |
| Recall@50 | 0.875 | 0.873 | 0.883 | 0.884 | 0.872 | 0.886 | 0.891 | 0.886 | 0.889 | **0.906** |

further support the robustness of our findings, emphasizing the consistency of the performance improvements achieved with an increased number of clusters up to a certain point.

## A.7 Latency and Performance Comparison

We show the efficiency comparison (with the performance comparison) across models in Table 9. For a fair comparison, we run all experiments on a single NVIDIA A100 GPU with TensorFlow framework (version 1.12) without any further optimization on the computation. We choose the Taobao dataset and set the batch as 1.

**Result and analysis.** In examining the efficiency of various recommendation models, notable differences in training and inference times highlight the diverse computational demands across these models. Models like YoutubeDNN and GRU4Rec present a more efficient profile, offering lower latency during both training and inference phases, while perform the worst on the retrieval and ranking tasks. The variability in model efficiency, as seen in the standard deviations of training and inference times, indicates a potential fluctuation in latency that could impact real-world deployment. Our proposed model, GPR4DUR, stands out with its superior performance on the retrieval task (e.g., measured by interest-wise relevance, IR@50) and the ranking task (e.g., measured by recall, Recall@50). This suggests that for applications valuing enhanced retrieval and ranking accuracy, the trade-off of increased latency is acceptable. Additionally, GPR4DUR exhibits a low standard deviation in its timing metrics. This consistency in processing times is advantageous for online deployment. It offers a predictable performance, which is crucial for developers when balancing between model efficiency and effectiveness.

Conclusively, GPR4DUR, along with the other baseline models evaluated, meets the latency requirements for online application without the need for additional computational and serving optimizations. While GPR4DUR exhibits a higher time cost relative to some baselines, its performance improvement offers a compelling advantage.

Table 10: Result comparison between GPR4DUR and generative recommendation methods that also consider uncertainty. Here we use the Amazon dataset as an example to report the results and we confirm the observation is similar on the other two datasets. ⇑ indicates the higher the better; ⇓ indicates the lower the better.

| Model | Retrieval Task | | | | Ranking Task | |
|---|---|---|---|---|---|---|
| | IC@50 ⇑ | IR@50 ⇑ | ED@50 ⇓ | TEI@50 ⇑ | Recall@50 ⇑ | nDCG@50 ⇑ |
| VAECF | 0.783 | 0.482 | 0.546 | -0.082 | **0.909** | 0.913 |
| DiffuRec | 0.834 | 0.540 | 0.442 | -0.057 | 0.892 | 0.916 |
| DREAM | 0.812 | 0.521 | 0.460 | -0.091 | 0.889 | 0.917 |
| GPR4DUR | **0.895** | **0.560** | **0.423** | **-0.039** | 0.908 | **0.922** |

## A.8 Comparison with Generative Recommendation Methods

We note that some generative recommendation methods also consider user uncertainty, making them potential candidates for comparison with our method. These methods include VAECF [54], which assumes a distribution over user/item representations, and more recently diffusion-based recommendation models, DiffuRec [55] and DREAM [56]. We do not compare them in the main paper due to *(i) different design purpose*: these methods are not designed for multi-interest retrieval, which is the focus of this work, and *(ii) different model family*: VAECF is a parametric model while our model is non-parametric, and diffusion-based methods require an entirely different training procedure and sampling strategies.

To still satisfy this curiosity, we present a comparison between GPR4DUR and the generative recommendation methods on the Amazon dataset for both the retrieval task and ranking task. As shown in Table 10, the results still confirm the effectiveness of our model.

## A.9 Limitations and Future Work

In Sec. 4.5, we discussed methods to reduce the time complexity of traditional Gaussian Process Regression (GPR). Despite these efforts, the training and inference times presented in Table 9 remain slightly higher than those of recent state-of-the-art methods. Further improving the efficiency of our approach would be an intriguing area for future research.

Moreover, the integration of collaborative Gaussian processes offers a promising avenue. Currently, our model focuses primarily on personalization, using other users only for tuning GPR hyperparameters. We hypothesize that leveraging collaborative learning techniques, such as those described in [57], could significantly enhance the performance and effectiveness of our method.

## A.10 Broader Impacts

The application of Gaussian Process Regression (GPR) for user modeling in recommendation systems has both positive and negative societal impacts. Below, we outline these impacts in detail.

**Positive Societal Impacts**. Our approach, GPR4DUR, aims to significantly enhance the personalization and efficiency of recommender systems. By effectively capturing the diverse and dynamic interests of users, GPR4DUR can improve user satisfaction and engagement across various online platforms. This could lead to more tailored content delivery, helping users find relevant and interesting content more quickly and reducing information overload. Furthermore, the uncertainty-aware nature of our method allows for better exploration-exploitation balance, potentially uncovering niche interests and underrepresented content that users might find valuable. This can promote diversity and inclusivity in content consumption, providing a wider range of options to users.

**Negative Societal Impacts**. Despite these positive aspects, there are potential negative societal impacts associated with the deployment of our method. One major concern is the potential for reinforcing existing biases in recommendation systems. If the training data reflects societal biases, GPR4DUR might inadvertently perpetuate these biases, leading to unfair treatment of certain user groups. Additionally, the improved personalization capabilities might increase the risk of creating filter bubbles, where users are only exposed to content that reinforces their existing preferences, limiting their exposure to diverse viewpoints.

Another potential issue is privacy. As GPR4DUR leverages detailed user interaction histories to model preferences, there is a risk of sensitive information being inferred or misused. Ensuring robust data protection and adhering to privacy standards is crucial to mitigate this risk.

**Mitigation Strategies**. To address these potential negative impacts, several mitigation strategies can be employed. Firstly, implementing fairness-aware algorithms that explicitly account for and mitigate biases during model training can help ensure equitable recommendations across different user groups. Secondly, incorporating mechanisms to introduce serendipity in recommendations can counteract filter bubble effects, exposing users to a broader range of content. Lastly, adhering to strict data privacy regulations and employing advanced anonymization techniques can safeguard user data and privacy.

Overall, while GPR4DUR has the potential to significantly improve user modeling and personalization in recommender systems, careful consideration and implementation of mitigation strategies are essential to address its broader societal impacts.

