# OpenReview forum: "Density-based User Representation using Gaussian Process Regression for Multi-interest Personalized Retrieval"
_NeurIPS.cc/2024/Conference — NeurIPS 2024 poster_

### Official Review · Reviewer_icLv · 2024-07-07

**Soundness:** 3
**Presentation:** 3
**Contribution:** 3
**Rating:** 6
**Confidence:** 3

**Summary:**

This work presents a novel method for user representation in retrieval and recommendation systems. The authors propose a density-based approach that leverages the density of user interactions to enhance the representation of users. This method is aimed at improving the performance of retrieval and recommendation systems by better capturing the underlying user preferences and behaviors. The paper includes detailed experiments and evaluations demonstrating the effectiveness of the proposed approach compared to existing methods.

**Strengths:**

S1. The paper introduces a unique density-based user representation method that is novel in its approach. This originality is reflected in the way the authors have redefined user representation by focusing on the density of user interactions, which is a creative and fresh perspective in the field of recommendation systems.

S2. The quality of the research is high, as evidenced by the thorough experimentation and detailed evaluations. The authors have conducted extensive tests to validate their method, comparing it with several baselines and showing significant improvements in performance.

S3. The paper is well-written and clearly structured. The authors have provided comprehensive explanations of their methodology, including detailed mathematical formulations and algorithmic steps. The experimental results are presented in a clear and understandable manner.

**Weaknesses:**

W1. The proposed method uses item categories as auxiliary information, while most of the baselines only consider user-item interaction information. This may lead to unfair comparisons.

W2. Some important baselines are missing. Specifically, STOSA[1], DT4Rec[2], and BERD+[3] use Gaussian distribution to model the uncertainty of user preferences in sequential recommendation, which are highly relevant to this work.

W3. The Interest-wise Relevance (IR) metric might not be reliable, as it relies on the learned item embeddings which may be inaccurate.

W4. The manuscript contains some typos. For example, on line 124, 'a RS' should be 'an RS'. On line 155, there should be an 'is' before the covariance function.



References:

[1] 	Ziwei Fan, Zhiwei Liu, Yu Wang, Alice Wang, Zahra Nazari, Lei Zheng, Hao Peng, Philip S. Yu. Sequential Recommendation via Stochastic Self-Attention. WWW 2022: 2036-2047.

[2] Ziwei Fan, Zhiwei Liu, Shen Wang, Lei Zheng, Philip S. Yu. Modeling Sequences as Distributions with Uncertainty for Sequential Recommendation. CIKM 2021: 3019-3023.

[3] Yatong Sun, Xiaochun Yang, Zhu Sun, Bin Wang. BERD+: A Generic Sequential Recommendation Framework by Eliminating Unreliable Data with Item- and Attribute-level Signals. ACM Trans. Inf. Syst. 42(2): 41:1-41:33 (2024).

**Questions:**

Please refer to the Weaknesses section.

**Limitations:**

The authors have acknowledged several limitations in their work, especially the potential challenges in scalability.

---

> ### Author Rebuttal · Authors · 2024-08-06
>
> Thank you for your positive feedback and appreciation of our method's novelty, soundness, and presentation. We address your main concerns below.
>
> **W1 Clarification on using the item categories**
>
> We would like to clarify that many strong baselines we are comparing use the category information, such as MIND, ComiRec, and CAMI (short for Category-Aware Multi-Interest), thus we believe our comparison is fair. We avoid modifying other baselines which do not leverage category information. We view this as an inherent limitation of these approaches, as such information is fruitful and often easily available in real-world systems.
>
> Additionally, we emphasize that using category information during item pretraining is necessary, as shown in Figure 6, Appendix A.3. Pretraining without the item-category auxiliary loss in Eq. (5) leads to misaligned item embeddings.
>
> **W2 Clarification on baselines**
>
> We acknowledge that the cited works are relevant to user modeling in terms of capturing uncertainty. We appreciate the reviewer for pointing out these works and will add them in the related work section in our final manuscript.
>
> Our task is multi-interest retrieval (lines 115-116, lines 139-142), thus most state-of-the-art baselines we are comparing to are designed for this task, such as all MUR baselines. The focus of the works pointed out by the reviewer differs somewhat from ours: they each focus on the ranking phase rather than the retrieval phase (e.g., paper [1] and [2] pointed out by the reviewer use cut-off relevance metrics merely at lengths one and five to measure the ranking performance), and none of them have the goal of finding a collection of items that can cover a user’s multiple interests. Additionally, our method is non-parametric without additional transformer layers or feed-forward layers, which is also a big difference in the model architecture compared to the works pointed out by the reviewer. Given the 11 baselines in our experiment section, the 3 generative recommendation methods (including some very recent ones) in Table 10, Appendix A.8, and the different focus of the work pointed out by the reviewer, we believe our experiments convincingly show the effectiveness and efficiency of our method.
>
> **W3 Reliability of the IR metric**
>
> Thanks for pointing this out: we considered this during our design and will make this more explicit in the discussion. As shown in lines 271-273, *“To obtain ground-truth similarities between items, and to mitigate the influence of the chosen pre-trained model, we pretrain the item embeddings using YoutubeDNN with a higher dimension size (d = 256) …”* Considering that YoutubeDNN is a mature method that has shown its success and reliability in one of the largest real-world recommendation platforms, we think it is reasonable to pretrain YoutubeDNN with a high dimension size to obtain reliable item embeddings. Our results on other metrics (IC, ED, TEI, Recall, nDCG) also demonstrate the superiority of our method over prior baselines on the multi-interest retrieval, showing a similar trend as the IR metric.

---

> > ### Comment · Reviewer_icLv · 2024-08-13
> >
> > Thanks for the authors' explanations. I have no further questions.

---

> > > ### Author Response · Authors · 2024-08-13
> > >
> > > Thank you for reading our rebuttal and for your thoughtful consideration. We are pleased that our responses addressed your concerns. We are happy to clarify any points in the remaining discussion window should you have additional questions.

---

### Official Review · Reviewer_Eeve · 2024-07-09

**Soundness:** 2
**Presentation:** 2
**Contribution:** 2
**Rating:** 5
**Confidence:** 4

**Summary:**

The paper introduces density-based user representations (DURs) using Gaussian process regression (GPR) to address limitations in existing user modeling methods for personalized recommender systems. The proposed GPR4DUR approach effectively captures user interest variability, incorporates uncertainty-awareness, and scales well, as demonstrated by experiments with realworld datasets and simulated users. This method balances exploration and exploitation, enhancing multi-interest recommendations without manual tuning.

**Strengths:**

1 The authors present a density-based approach for representing users in personalized multi-interest retrieval, incorporating Gaussian process regression for more sophisticated user modeling.

2 The paper introduces four novel evaluation protocols tailored for the multi-interest retrieval task.

3 The proposed method, GPR4DUR, is thoroughly validated through extensive experimentation on real-world datasets, showcasing its efficacy in the field.

**Weaknesses:**

1 The difference between SUR and MUR is not clearly explained. Why MUR is better, is there a reasonable explanation or proof? The author should explain this in detail in the introduction section.

2 The adaptability and uncertainty that the author repeatedly emphasizes, what is the specific manifestation in the recommendation scenario. Whether their modeling of user representation is meaningful, they should provide specific definitions, clear assumptions and more specific explanations.

3 The core motivation of the paper is not clear. For example, in line 33, the author did not explain why manually selecting clusters will limit the adaptability of MUR. How to understand the variability of interests across users? Why this type of method cannot simulate uncertainty well?

4 The paper intro is vague and abstract, and it is hard for readers to establish a connection between the motivation and the proposed method.

**Questions:**

1 For some simple collaborative signal modeling, can your method also achieve good performance improvement on it?

2 Is there a more detailed definition or explanation of density-based?

3 Figure 1 is not well explained. What do the x-axis and y-axis represent? How to understand the value range? Is normalization performed? The SUR and MUR graphs look the same. What do the color of the points mean?

4 The explanation on line 35 is too abstract. How to understand since interests generally have high variability across users. Is there a strict mathematical definition to explain this problem?

5 Lines 40-42, this three words are too abstract. Is there a more detailed explanation or explanation in mathematical formulas? Why can Gaussian process solve these three points?

6 Line 36, why is uncertainty regarding a user’s interests not well-modeled by these methods, diminishing their ability to perform effective online exploration.

**Limitations:**

The core motivation is not clearly explained, and it is difficult for the author to understand how the proposed method solves the problems of existing methods. This paper is not recommended for acceptance.

---

> ### Author Rebuttal · Authors · 2024-08-06
>
> Thank you for taking the time to review our paper. We address your main concerns below.
>
> **W1+Q2+Q3 Clarification on the difference of SUR, MUR, DUR**
>
> SUR uses a single point (K=1) in the embedding space to represent a user, while MUR uses multiple points (K>1, e.g., K=4 in MaxMF) to represent a user. We believe this distinction is clearly explained in the introduction and also in the name of these two categories (lines 24-26, 30-32). MUR generally outperforms SUR because users often have multiple interests, which are better captured by multiple points in the embedding space. This has been shown by previous works on MUR which we cited. For example, a user interested in documentary, horror, action, and comedy movies is more accurately represented by multiple points, each reflecting a different interest, as mentioned in line 31.
>
> Regarding Figure 1, lines 67-70 and the figure caption clarify that it is a t-SNE visualization of the prediction scores between a user and all items, reduced from an original embedding size (i.e., 64) to 2D. The x-axis and y-axis do not have specific meanings but form a 2D embedding space. The score, defined as *“the inner-product between user and item embeddings”* (lines 67-68), is represented by the color bar in each subfigure, with more “red” indicating higher scores and more “blue” indicating lower scores. While the subfigures for SUR and MUR look similar because both are point-based methods, the high-score region (red) in MUR is larger than in SUR, covering more small triangles (as described in line 70), indicating that MUR captures more user interests.
>
> For DUR (Density-based User Representation), it uses a function over the item embedding space to encode user preferences (lines 45-47). The mathematical definition of DUR is provided in Section 4.1.
>
> **W4+Q5 Clarification on adaptability, uncertainty-awareness, efficiency**
>
>  - **W2+W3+Q4 - *Adaptability***
>
> In real-world systems, users have varying interests. For example, some users only like comedy movies, while others have broader interests. A user may also have different preferences depending on their mood and other contexts. This variability is what we refer to as "interests generally have high variability across users" (lines 35-36). This is also verified in our datasets, and we provide an analysis in "Performance across User Groups" (lines 350-360), where users are grouped based on the number of interests. Therefore, manually choosing a constant number of clusters (K) across all users is suboptimal. Deriving a personalized K is also hard since we do not know the exact number of interests a user may have, and new interests may emerge.
>
> We are unsure what the reviewer means regards to “the specific manifestation in the recommendation scenario” as well as the concerns on “whether their modeling of user representation is meaningful”. Our DUR method is well-defined and well-motivated to solve the limitations of prior methods for capturing user’s multiple interests with Gaussian process regression. Also, we evaluate our method using three widely used datasets, comparing with 11 baselines in Section 5 and three generative recommendation methods in Appendix A.8, across four retrieval metrics (Table 1) and two ranking metrics (Table 2). The consistent superiority of our method demonstrates its ability to capture diverse user interests and improve retrieval and recommendation quality. Thus, our method has both theoretical grounding and empirical support for the improvements it offers.
>
>  - **W2+Q6 - *Uncertainty-awareness***
>
> The SUR and MUR methods cited in paragraphs 2 and 3 of the introduction do not consider uncertainty in user preferences as part of their model design. We also describe this in the second paragraph of the related work (lines 105-110, Section 2), noting that these methods use a deterministic approach to compute the relevance score between a user and an item. Capturing uncertainty is crucial for online exploration in recommendation systems as it balances exploiting known preferences and exploring new items, leading to improved personalization and adaptability to changing preferences. The ability of our method to capture uncertainty comes from the inherent nature of Gaussian Processes.
>
> - **Q5 - *Efficiency***
>
> Efficiency, as mentioned in lines 43-44, relates to avoiding high-dimensional embeddings. Lines 362-369 and Figure 4 detail our experiments and analysis. GPR4DUR achieves competitive performance with low-dimensional item embeddings, while traditional methods perform poorly with low dimensions.
>
> **Q1 Clarification on “simple collaborative signal modeling”**
>
> We are unsure what “simple collaborative signal modeling” refers to. If the reviewer is referring to methods like Matrix Factorization, we highlight that our comparisons include 11 baselines (Section 5.4), most of which do use collaborative filtering signals and are more powerful than Matrix Factorization (for example, YoutubeDNN). The comparison with these stronger and state-of-the-art methods demonstrates the effectiveness and efficiency of our approach. If the reviewer has something else in mind, we would be happy to respond to any clarification.

---

> > ### Comment · Reviewer_Eeve · 2024-08-13
> >
> > Thanks for the authors' detailed responses. After reading their responses, I decided to raise my score since the authors have addressed some of my concerns.

---

> ### Author Response · Authors · 2024-08-13
>
> Thank you for reading our rebuttal and for your thoughtful consideration. We are pleased that our responses addressed your concerns. We are happy to clarify any points in the remaining discussion window should you have additional questions.

---

### Official Review · Reviewer_r5NL · 2024-07-13

**Soundness:** 3
**Presentation:** 3
**Contribution:** 3
**Rating:** 5
**Confidence:** 4

**Summary:**

The research problem addressed in this study revolves around the challenge of effectively capturing user interests to enhance the quality of personalized recommendations. Traditional methods often struggle to model users with diverse interests accurately, leading to suboptimal recommendations.

The authors emphasize the limitations of existing user modeling techniques, such as collaborative filtering and matrix factorization, which may not adequately capture the multi-interest nature of users. To address this gap, the paper introduces the concept of Density-based User Representations (DURs) and proposes the GPR4DUR approach, which leverages Gaussian Process Regression to create more nuanced user representations based on the density of user interactions.

**Strengths:**

1.	Introducing the concept of Density-based User Representations (DURs) as a more effective way to capture user interests.
2.	Proposing the GPR4DUR approach, which utilizes Gaussian Process Regression to create personalized user representations based on interaction density.

**Weaknesses:**

1. A clearer articulation of the specific advancements or enhancements that Gaussian Process Regression for Density-based User Representation (GPR4DUR) provides over traditional methods—particularly in terms of adaptability.

2. Discussing the scalability of the GPR4DUR approach with larger datasets and its computational efficiency would help to assess the practical applicability of the technique.

**Questions:**

Could the authors provide more details on the selection criteria for the real-world offline datasets used in the experiments? What specific characteristics were considered in choosing these datasets, and how do these characteristics make them particularly suitable for evaluating the effectiveness of the GPR4DUR approach?

In the online experiments involving simulated users, what specific metrics were employed to assess the effectiveness of the GPR4DUR recommendation policies? Can the authors detail the methodology used to simulate user interactions and describe how the performance of GPR4DUR was evaluated in this online setting?

Considering the emphasis on scalability, how does the computational complexity of GPR4DUR compare to traditional user modeling methods as the number of users and items increases? Have there been specific experiments conducted to analyze the scalability of GPR4DUR across different dataset sizes? What were the findings regarding the impact of increasing data volume on system performance?

How does GPR4DUR address and mitigate biases during the model training process to ensure equitable recommendations across diverse user groups? Can the authors elaborate on the specific mechanisms or algorithms implemented within GPR4DUR to promote fairness and reduce potential biases in the recommendation outcomes?

**Limitations:**

The authors have adequately addressed the limitations and identified no potential negative societal impacts of their work.

---

> ### Author Rebuttal · Authors · 2024-08-06
>
> Thank you for your positive feedback and appreciation of our method's novelty, soundness, and presentation. We address your main concerns below and will add details and clarifications in the paper as needed.
>
> **W1 The advancements of GPR4DUR over prior baselines**
>
> Our proposed method, GPR4DUR, advances traditional methods in all three aspects described in the introduction: (1) *Adaptability*, (2) *Uncertainty-awareness*, and (3) *Efficiency*. We elaborate on these below and will update the discussion in the paper to make these contributions more evident.
>
>  - *Adaptability*: To demonstrate our method's adaptability to different user interest patterns, we evaluated it on two interest-wise metrics and two exposure-related metrics for the retrieval phase (Table 1) and two classical relevance metrics for the ranking phase (Table 2) across three widely used datasets. Our results indicate that, compared to prior SOTA MUR methods that use multiple points to represent a user, our GPR-based method can better capture user interests with a higher coverage ratio and retrieve more relevant items. This demonstrates that our method is more adaptable to different users regardless of their interest patterns.
>
>  - *Uncertainty-awareness*: Section 6 details our experiments and analysis, showing that GPR4DUR can effectively capture users' dynamic interests in an online setting. We highlight that traditional heuristic, SUR, and MUR methods (baselines in Section 5.4) do not model uncertainty and always use a deterministic approach to compute the relevance score between a user and an item. The ability of our method to capture uncertainty stems from the inherent nature of Gaussian Processes.
>
>  - *Efficiency*: Lines 362-369 and Figure 4 provide the experiment and analysis. GPR4DUR achieves competitive performance with low-dimensional item embeddings, while traditional methods perform poorly with low dimensions.
>
> **W2+Q1+Q3 Datasets and computational efficiency**
>
> We selected the three datasets due to their varying size and sparsity, different domains, and widespread adoption in the recommendation community, ensuring that our empirical results are generalizable. The Taobao dataset, containing 70 million user-item interactions, is considered large in the community. The dataset statistics are shown in Table 6, Appendix A.4. In general, our method consistently achieves better performance compared to baselines across different datasets as shown in Section 5.
> Computational efficiency is discussed in Section 4.5 where we provide empirical results in Table 9, Appendix A.7. In summary, GPR4DUR, along with the other baseline models evaluated, meets the latency requirements for real-world applications without needing additional computational and serving optimizations. While GPR4DUR exhibits a higher time cost than some baselines, its performance improvement offers a compelling advantage. The dataset used in this experiment is the largest one in this paper, i.e., the Taobao dataset. We acknowledge that further improving the efficiency of our approach would be an intriguing area for future research (see discussion in Section 4.5).
>
> **Q2 Clarification on the online simulation**
>
> The metric we report is interest coverage (see lines 397-399 and the caption of Table 3), which is the same as the interest-wise coverage (IC) (see line 262). We will better clarify this in the final manuscript. To report the values in Table 3, we compute a “cumulative” version of interest coverage by reporting the IC@10 (i.e., line 406, we recommend the top-10 items per user at each iteration) averaged across all users on all previously recommended items prior to the current iteration. Thus, Table 3 shows that using our proposed method with Thompson sampling, an average of 98% of interests can be covered for each user after 10 iterations in this online simulation experiment, outperforming methods without uncertainty.
>
> For the method used to simulate user interactions, we adopt the method cited in line 380: *“... we follow [46], first running a Markov Chain using a predefined user interest transition matrix to obtain the user’s interest interactions for S = 10 steps.”* Due to character restrictions, it is challenging to describe the full process in detail here. We refer the reviewer to Section 3.2 and Appendix A of the paper [46] for a comprehensive explanation. We will add more exposition of this method in the Appendix.
>
> [46]: Mehta et al. Density Weighting for Multi-Interest Personalized Recommendation.
>
> **Q4 Mechanisms to promote fairness and reduce bias**
>
> The ability of our proposed GPR4DUR to promote fairness and reduce potential biases is due to the inherent nature of Gaussian Process Regression (GPR) for capturing uncertainty. For items with fewer observations, such as items corresponding to niche interests interacted with by few users, the mean prediction will be close to the prior while the variance (uncertainty) will be high (see the toy example in Figure 2). Thus, when using UCB to sample items to generate the retrieval list (Section 4.2), items of niche interests still have a chance to be selected. We thus observe that our method has a low value on Exposure Deviation and a high value on Tail Exposure Improvement. Moreover, this “fairness” in the item exposure can also promote “fairness” in user satisfaction, since those users who generally like niche interests have a higher chance of finding the items they prefer.

---

### Official Review · Reviewer_F9WH · 2024-07-13

**Soundness:** 3
**Presentation:** 4
**Contribution:** 4
**Rating:** 8
**Confidence:** 4

**Summary:**

The authors propose GPR4DUR, which incorporates uncertainty-awareness and scales well to large numbers of users.  This method leverages Gaussian process regression (GPR) to effectively capture the diverse and dynamic interests of users without the need for manual tuning. Through experiments with real-world offline datasets and online simulations, the authors demonstrate that GPR4DUR can adapt to user interest variability and efficiently balance the exploration-exploitation trade-off by utilizing model uncertainty. The paper also presents new evaluation protocols and metrics for multi-interest retrieval, showing the adaptability and efficiency of GPR4DUR in capturing a user's multiple interests and recommending items that span those interests, including niche ones.

**Strengths:**

- The authors ingeniously apply Gaussian Processes (GP) to characterize the diversity and density of user interests, incorporating a measure of uncertainty that proves especially efficacious in identifying and capturing niche user interests with precision.

- The researchers have designed a comprehensive suite of evaluation metrics tailored for multi-interest contexts. These metrics, which encompass interest matching, interest coverage, and multi-interest balance, are poised to serve as a robust framework for guiding future research endeavors in this domain.

- The authors have validated the outstanding performance of GPR4DUR in both offline and online settings.

**Weaknesses:**

- As mentioned in section 4.5 of the document, GPR4DUR is more suited for the ranking stage of large-scale recommendation systems. However, the ranking stage often involves a variety of user features, such as user embeddings, to achieve more accurate ranking results. Yet, this paper only utilizes the user's historical interactions.

- The article still lacks an analysis of the relationship between multiple interests and uncertainty.

**Questions:**

- Is there greater uncertainty associated with a user's niche interests? Or, how does the uncertainty differ between items of varying popularity and users with different interaction patterns?

- In the tuning of GPR parameters, would using different parameters for different users yield better results?

**Limitations:**

Yes

---

> ### Author Rebuttal · Authors · 2024-08-06
>
> Thank you for your positive feedback and appreciation of our method's novelty and in-depth analysis. We address your main concerns below.
>
> **W1 Clarification on the focus of our work**
>
> As mentioned in our paper (lines 119, 139, 212-213), our approach focuses on the retrieval phase and is best suited to scenarios with millions or tens of millions of items, of which O(100) items are retrieved for downstream ranking. The ranking stage is not our focus. Rich user features can be used in the ranking stage, as you rightly observe, to achieve more accurate ranking results if such auxiliary information is available in the dataset. We do note, however, that our method performs competitively in the ranking phase (Table 2), even though our main focus is on retrieval. Finally, user embeddings are also based on a user’s historical interactions, so our approach still uses the same raw information.
>
> **W2+Q1 Relationship between interests and uncertainty**
>
> Thanks for the question. If we are interpreting it correctly, for items with fewer observations, such as items in niche interests interacted with by few users, the mean prediction will be close to the prior while the variance (uncertainty) will be high (see the toy example in Figure 2). This is due to the nature of GPR. If we misinterpreted your question, please let us know.
>
> **Q2 GPR parameter tuning**
>
> It is indeed possible to tune the GPR parameters for each user, but this requires a holdout set with sufficient data per user. In our experiments, we have no more than 160 interactions per user, so we leverage collaborative information for tuning the hyperparameters of the GPR.

---

### Decision · Program_Chairs · 2024-09-25

**Decision:**

Accept (poster)

**Comment:**

The paper proposes a method called GPR4DUR for multi-interest recommendation and retrieval by using Gaussian process regression and density-based user representation.

Reviewers mostly agree on the following contribution of the paper:
* using Gaussian Processes (GP) and density-based user representation to capture multi-interest of users and providing a measure of uncertainty on the recommendation is novel (all reviewers),
* performance study is comprehensive (F9WH, Eeve, icLv)
* design new evaluation metrics tailored for multi-interest contexts (F9WH, Eeve)

The reviewers raised questions on scalability, clarity on adaptability and comparing with more relevant baselines etc.. Some of them are addressed during rebuttal. The authors are advised to incorporate these additional analysis and results in the final version.